# Computational structural studies of SGLT2-related polypharmacy

**Aled Lloyd** *, **Karl Austin-Muttitt, Jonathan G.L. Mullins**

Structural Bioinformatics Group, Faculty of Medicine, Health and Life Science, Swansea University, Wales, United Kingdom

* 360839@swansea.ac.uk

## Abstract

### Introduction

Sodium Glucose Cotransporter 2 (SGLT2) is the main active transport protein involved in sodium and glucose reabsorption in the kidney. SGLT2 inhibitors (SGLT2i) are widely recommended for patients with diabetes, heart failure and chronic kidney disease (CKD). Many multimorbid patients are prescribed these compounds, raising questions about polypharmacy. We have performed a computational drug repurposing screen to identify other licensed drugs capable of binding at or near the SGLT2i active site aiming to identify compounds that could either compete with SGLT2i or inhibit sodium and glucose transport.

### Methods

The library of BNF listed compounds was obtained from NCBI PubChem. D-I-TASSER was used to generate monomeric structural models, and MODELLER was used to incorporate MAP-17 and empagliflozin from a reference structure (PDB 7VSI). CHARM-GUI was used to insert the protein into a membrane. The structural model was refined in a 5 nanosecond GROMACS equilibration. Docking studies using PLANTS were performed and compounds interacting with key protein residues were identified. CHARM-GUI was used to prepare membrane- and ligand-bound systems to run in GROMACS using GPUs in Google Colab. 10 nanosecond simulations were undertaken (300 Kelvin and 1 bar) to discriminate between binding and non-binding events.

### Results

The SGLT2-MAP17 structure was obtained in the inward-open conformation, showing good agreement with published structures. Existing SGLT2i (empagliflozin, dapagliflozin, canagliflozin, ertugliflozin) all feature in the top 1% of docked compounds in the repurposing screen. 17 compounds were investigated by MD, with all of them remaining bound to the protein in simulation. Ceftriaxone, tobramycin, clindamycin,

**Data availability statement:** The structural models, docking results, molecular dynamics simulation results and input files generated and analysed in this study are available at the figshare data repository using the DOI https://doi.org/10.6084/m9.figshare.28363397.v1.

**Funding:** The author(s) received no specific funding for this work.

**Competing interests:** The authors have declared that no competing interests exist.

fluvastatin, atorvastatin and ticagrelor were among the compounds with potentially significant interactions.

## Discussion

It is not clear whether the stable ligand interactions identified here would result in inhibition of sodium and glucose transport, or if the interactions could provide competitive inhibition for SGLT2i compounds currently used. The compounds identified are not presently recognised as interacting with SGLT2i, nor are they associated with any adverse effects suggesting inhibition of the protein. This study is limited by considering only the protein-ligand interaction and not wider pharmacokinetic or pharmacodynamic factors.

## Conclusion

The indication of interactions with several compounds likely to be prescribed alongside SGLT2 inhibitors, such as antibiotics, statins, and antiplatelet agents, warrants further investigation of the potential for polypharmacological complications.

## Introduction

Sodium Glucose Cotransporter 2 (SGLT2) is the main protein involved in sodium and glucose reabsorption in the kidney [1]. Glucose transport through SGLT2 is enhanced 100-fold by the presence of MAP17, a membrane protein considered an essential subunit of SGLT2 [2]. Substrate transport occurs in three steps; first sodium ions bind to the extracellular region of SGLT2, opening the glucose binding domain [1]. Once extracellular glucose is captured the protein flips to an inward open position allowing release of both sodium and glucose intracellularly The protein then reverts to its original position [1]. While sodium and glucose transport through SGLT2 is not an active process, maintaining the intracellular sodium concentration as a result of SGLT2 transport expends a significant amount of ATP, estimated to be 12% of the total renal ATP expenditure [1]. The structure and mechanism of inhibition by gliflozin-based SGLT2 inhibitors (SGLT2i) has been elucidated [3]. These compounds have a sugar-like end and this binds to the glucose domain while the protein is in its inward open conformation, keeping the protein in that conformation [3]. The chemical structures of a selection of licensed SGLT2i can be found in Fig 1. Inhibition of SGLT2 results in heavy glycosuria and natriuresis. These effects cause a reduction in blood pressure and plasma volume and increase lipid utilisation and ketogenesis [1,4]. Insulin, glucagon and intestinal hormones are all mediates as a result of the metabolic changes [1]. The degree of glycosuria is related to the cardiorenal benefits of SGLT2 inhibition with greater benefits observed in patients with heavier glycosuria [5].

SGLT2 requires little introduction in the fields of diabetes, nephrology and cardiology as inhibitors of this protein are mainstays in the treatment of several common conditions based on impressive clinical trial results. Most type 2 diabetes mellitus (T2DM)

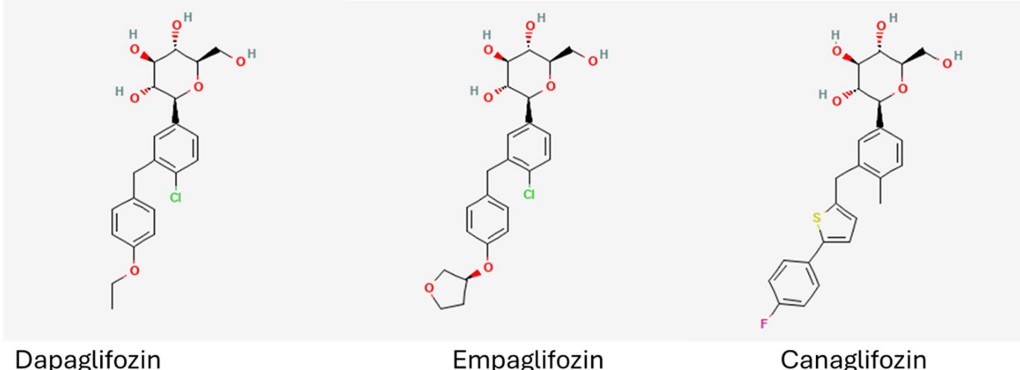

Dapaglifozin    Empaglifozin    Canaglifozin

**Fig 1. The chemical structure of dapagliflozin, empagliflozin and canagliflozin obtained from the NCBI PubChem database [6].**

management guidelines advocate the use of SGLT2i as part of combination therapy early in the course of the disease [7]. Prescription of one of these agents in diabetic patients reduce glycosylated haemoglobin, weight and microalbuminuria in patients with T2DM [8,9]. Cadiovascular studies were initially a regulatory safety requirement but cardiovascular protective effects were observed during the development of SGLT2i(10). The evidence for these benefits cumulated in clinical trial and meta-analysis data finding that major adverse cardiovascular events were reduced by 11% in patients with heart failure taking an SGLT2i. Hospitalisation and progression of renal disease was also reduced in these patients [10–12]. Based on these benefits, SGLT2i are recommended for the treatment of heart failure [13,14]. Large randomised control trials found a reduction in CKD progression, proteinuria and reduced all-cause mortality in patients with CKD receiving SGLT2 inhibitors [15,16]. As a result, SGLT2i are now recommended for patients with CKD [17,18].

The population risk of polypharmacy is likely to increase because of the recommendations to use SGLT2i in common chronic conditions. Polypharmacy is generally considered to be the prescription of 5 or more medication and is more likely in patients with more comorbidities [19]. The probability of adverse events are increased in older persons with polypharmacy [20]. These adverse effects could either take the form of unintended side effects or reduced clinical effectiveness of one or more of the prescribed drugs.

Given recommendations to use SGLT2i to treat common chronic conditions; and as patient groups are in general ageing and becoming more comorbid, there is significant potential for drug interactions caused by polypharmacy. We explored drug binding near the SGLT2i binding site using a virtual library screen of licensed medicinal compounds to identify the potential for competitive inhibition of SGLT2i or previously unrealised inhibition of glucose transport.

This study is the second study performed as part of the development of a study protocol for structural modelling-docking-molecular dynamics studies using automated web tools with graphical user interfaces. The aim of this protocol is to make structural bioinformatics studies accessible and achievable for new researchers to the field. Consideration therefore during the design of this study has been given to reducing the complexity and resource burden of performing and analysing each step of the experiment. Methodologically this study builds on the first study with this protocol, a virtual library screen of aquaporin 1 [21], by using MD to prepare the structural model for docking studies and by analysing the median root mean square deviation from three MD simulations ten nanoseconds in length to assess docked conformation stability.

## Materials and methods

### Structural modelling

The crystal structure of the SGLT2-MAP17 complex with bound empagliflozin has been resolved (PDB: 7VSI). However, this structure did not incorporate a long flexible segment, residues 571–654 [3]. Structural modelling using

D-I-TASSER [22] for the complete monomeric protein structure necessary for MD studies was performed. This was followed by homology modelling using MODELLER [23,24] to incorporate MAP17 and empagliflozin cofactors using PDB structure 7VSI as a scaffold and the Na$^+$ position was incorporated from a homologous LeuT scaffold [3,25]. In order to remove experimental artefact a long GROMACS molecular dynamics equilibration was undertaken. The centroid of a 5 nanosecond GROMACS molecular dynamics equilibration was used for docking studies. The protein model was inserted into a POPC phospholipid bilayer using the membrane builder facility of CHARMM-GUI prior to docking studies [26,27].

## Docking

To determine which docking method and search strategy would provide the best fit for this protein, each method and approach was tested using the location of empagliflozin in the PBD 7VSI structure. Five different docking methods and two different approaches (using a reference structure, and a search radius) were implemented. The approach resulting with the lowest RMSD, according to DockRMSD [28], when compared with the reference empagliflozin conformation of the PDB 7VSI structure [3], was used as the central benchmark for the repurposing screen.

The library of compounds for the repurposing screen using the best performing docking approach, comprised of 1512 compounds listed in the British National Formulary (BNF) [29]. Automated key residue analysis was then performed using an in-house script that provides a score based on hydrogen bonds or van der Waals contacts between the ligand and specified residues. Residues that interact with empagliflozin in PDB 7VSI through close contact or by hydrogen bonding were used for this analysis. Close contact was defined as a distance of 5Å, and the residues within 5Å of empagliflozin in the reference structure [3] were ASN 75, HIS 80, THR 87, PHE 98, GLU 99, SER 287, TYR 290, LYS 321, and GLN 457. These residues are labelled in Fig 2. A combination of calculated binding energy and proportion of key residue overlap was used to select clinically relevant compounds for studying using MD.

## Molecular dynamics

We have developed a bespoke cloud based system for preparing MD input files using CHARMM-GUI [30–34] and running MD simulations using GROMACS [35,36] entirely on graphics processing units in Google Colaboratory [37]. This system called GROMACS-on-Colab is freely available to use through Github at https://github.com/bioinfkaustin/gromacs-on-colab. This system has a facility for inserting ligand poses obtained from docking studies. 10 ns simulations were carried out for each compound with three simulations for each compound. While short by the compared to most MD studies this study duration was a considered decision. MD studies as a tool to assess the stability of docking study results have been validated in a study that used RMSD as the main measure [38]. In this study the difference in sensitivity measured by the area under a ROC curve (AUC) between each simulation length was relatively small. The calculated AUC values for experiments using the DUD-E dataset were 0.806, 0.840 and 0.836 for simulations of 10, 50 and 100 nanoseconds respectively [38]. From these figures it is apparent that a 50 nanosecond MD simulation confers only a modest improvement in sensitivity compared to a 10 nanosecond simulation. There is no benefit from a sensitivity perspective in a 100 nanosecond simulation compared to the 50 nanosecond simulation. The computational resources needed to run MD simulations are significant and could represent a barrier to new researchers to this field. Three simulations 10 nanoseconds were felt to give the best compromise between robust, reproducible simulations and rationing computational resources. Unstable ligand poses are likely to be expelled from the binding pocket relatively quickly; therefore, three separate shorter simulations were selected over a single longer simulation.

We are not aware of any validation studies for post-docking MD experiments that have used alternative methods of measurement such as RMSF, RG or MMPBSA analyses. Given the uncertainty of the relevance of these measurement methods they have been omitted from this study. Simplifying MD data analysis has the additional benefit of making the protocol under development easier to follow.

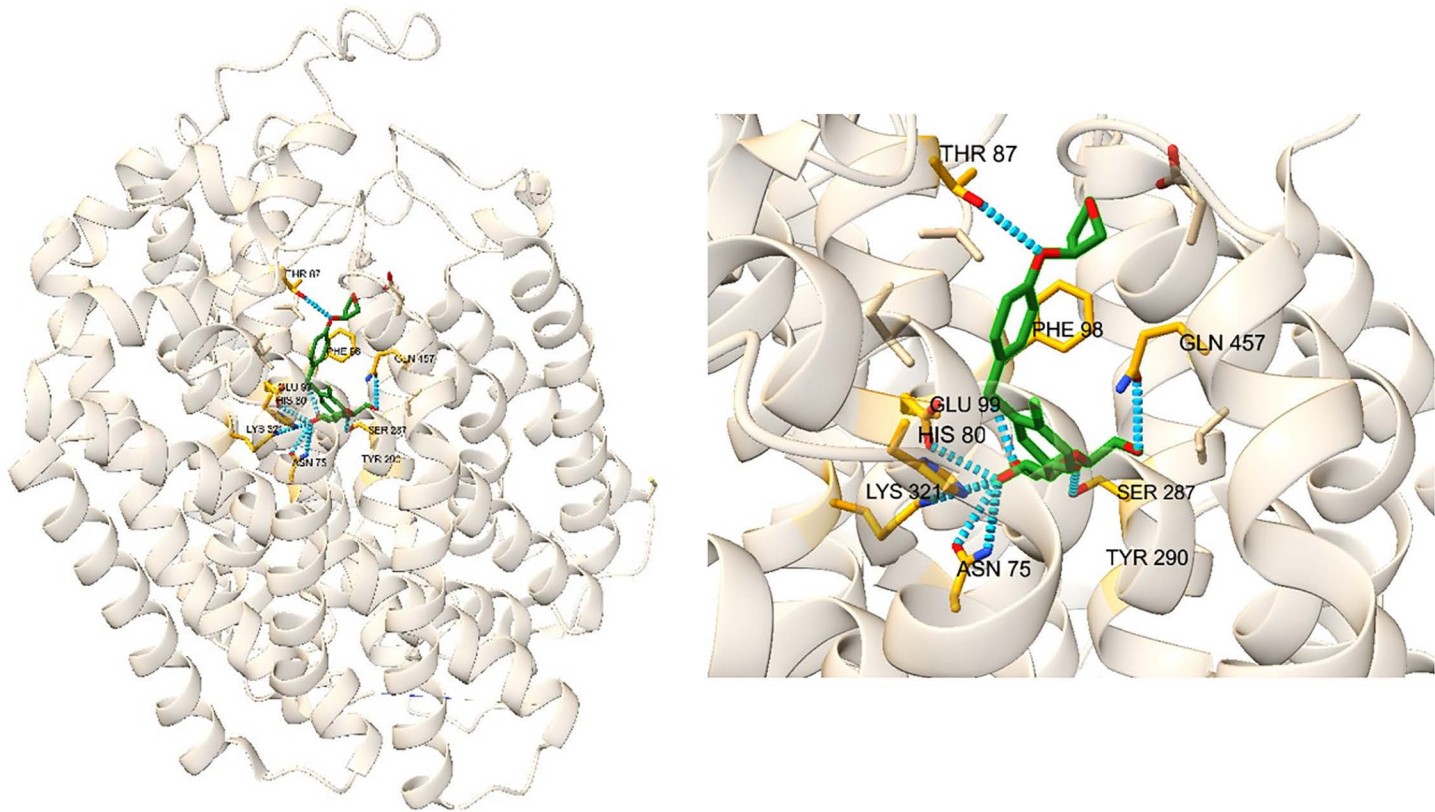

**Fig 2. SGLT2-MAP17 complex with empagliflozin (green) from PDB 7VSI [3].** Key residues for binding are labelled and highlighted in orange. Hydrogen bonds are denoted by the dotted blue lines.

A 5.5 Å RMSD early stop mechanism was used as RMSD values above this threshold are considered to be non-binding [38]. Any simulations exceeding this were assigned an RMSD of 10 Å for the purposes of calculating median RMSD values. We adopted the median RMSD between 9 and 10 ns to rank ligands.

## Results

### Structural modelling

The protein structure obtained in the inward-open conformation demonstrated good structural agreement with a published structure [3]. The position of the membrane was verified via alignment with the UniProt entry for SGLT2 (P31639) [39]. A structural overlay demonstrating the agreement between the model created and the reference structure published by Niu and colleagues(PDB 7VSI) [3] can be found in Fig 3. Most of the relatively small differences in these structures in Fig 3 can be related to the movement of side chains in the MD relaxation simulation the model built and used for these docking studies has been based upon. Comparison between these structures using the matchmaker tool in UCSF chimera identified RMSD values of 1.211 Å between 467 pruned atom pairs and 1.894 Å between all 586 pairs [40]. The position of the membrane was verified via alignment with the UniProt entry for SGLT2 (P31639) [39].

### Docking

We tested five different docking methods and two different approaches to define the search volume (using a reference structure, and a search radius) and used the approach with the lowest RMSD compared to the empagliflozin

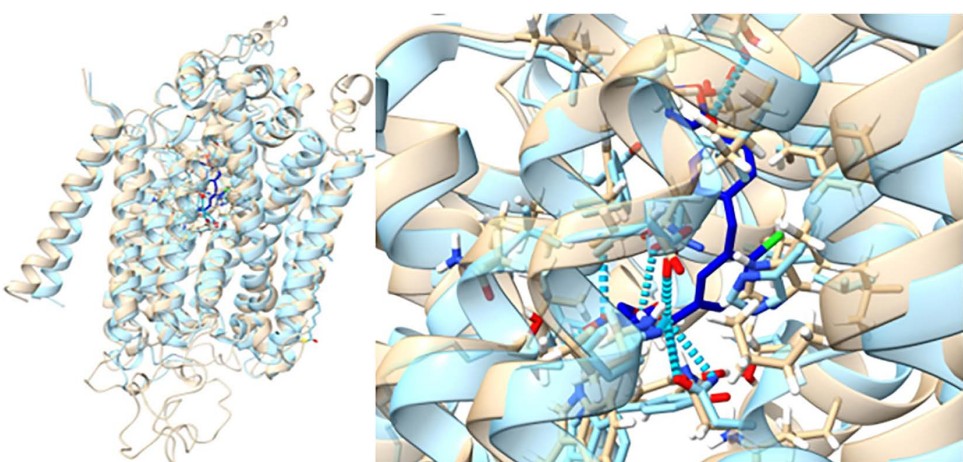

**Fig 3. The SGLT2-MAP17 complex model used for docking studies in this experiment (light brown) overlayed with the experimental structure PDB 7VSI (light blue) [3].** The empagliflozin molecule in the experimental structure is included and coloured royal blue.

conformation in the PDB 7VSI structure [3] according to DockRMSD [28]. As shown in Table 1, the PLP scoring system from PLANTS was used based on it achieving the lowest RMSD in the comparison with the reference structure. After key residue analysis, accounting for clinical indication and the PLANTS score, 17 candidate conformations were identified for testing using MD.

Fig 4 displays the process by which the compound with the best docking study results were selected for an assessment of stability using MD. The thresholds for key residue score and binding energy were estimated based on an assessment of the docking results of the known inhibitors. There is as a result, an inherent selection bias for larger compounds capable for forming multiple hydrogen bonds within the sample of compounds assessed by MD. The clinical assessment of drug characteristics focussed on the route of administration, frequency of prescription, and clinical indication. The aim of this assessment was to ensure that the compounds assessed by MD were systemically absorbed and frequently prescribed.

Existing inhibitors of SGLT2 (empagliflozin, dapagliflozin, canagliflozin, ertugliflozin) all feature in the top 1% of docked compounds in the repurposing screen. As the four known inhibitors were ranked in the top 1% of 877 docked compounds, an effective enrichment factor of $EF_{1\%} = 97.1$ was achieved by this docking stage. The known inhibitors were included in the sample tested using MD. A selection of the best scoring conformations, which were selected for MD studies, are depicted in Fig 5. The binding mode of each gliflozin compound is very similar, the sugar moiety sits deep within the pore supported by several hydrogen bonds. The remaining largely aromatic portions of these molecules sit near protein side chains creating a hydrophobic environment. Hydrophobic interactions appear to predominate in the interaction between SGLT2 and raltitrexed, where comparatively few hydrogen bonds can be seen in Fig 5.

**Table 1. RMSD values using different docking methods for empagliflozin docking to SGLT2 compared to PDB 7VSI.**

| Reference ligand method | | Box/radius method | |
|---|---|---|---|
| Docking method | RMSD (Å) | Docking method | RMSD (Å) |
| Chem PLP | 6.547 | Chem PLP | 0.775 |
| PLP | 1.037 | **PLP** | **0.765** |
| PLP95 | 0.979 | PLP95 | 1.004 |
| Autodock Vina | 0.846 | Autodock Vina | 0.795 |
| Vinardo | 0.918 | Vinardo | 0.969 |

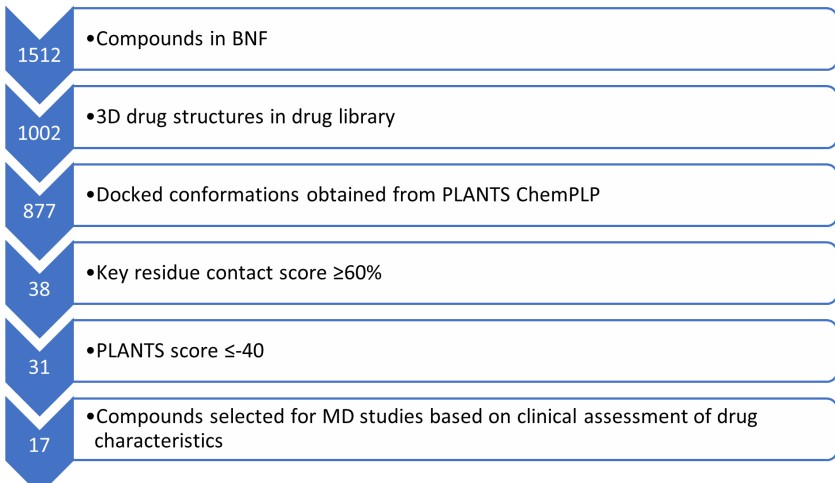

**Fig 4. Flowchart of MD candidate ligand selection from docking results.**

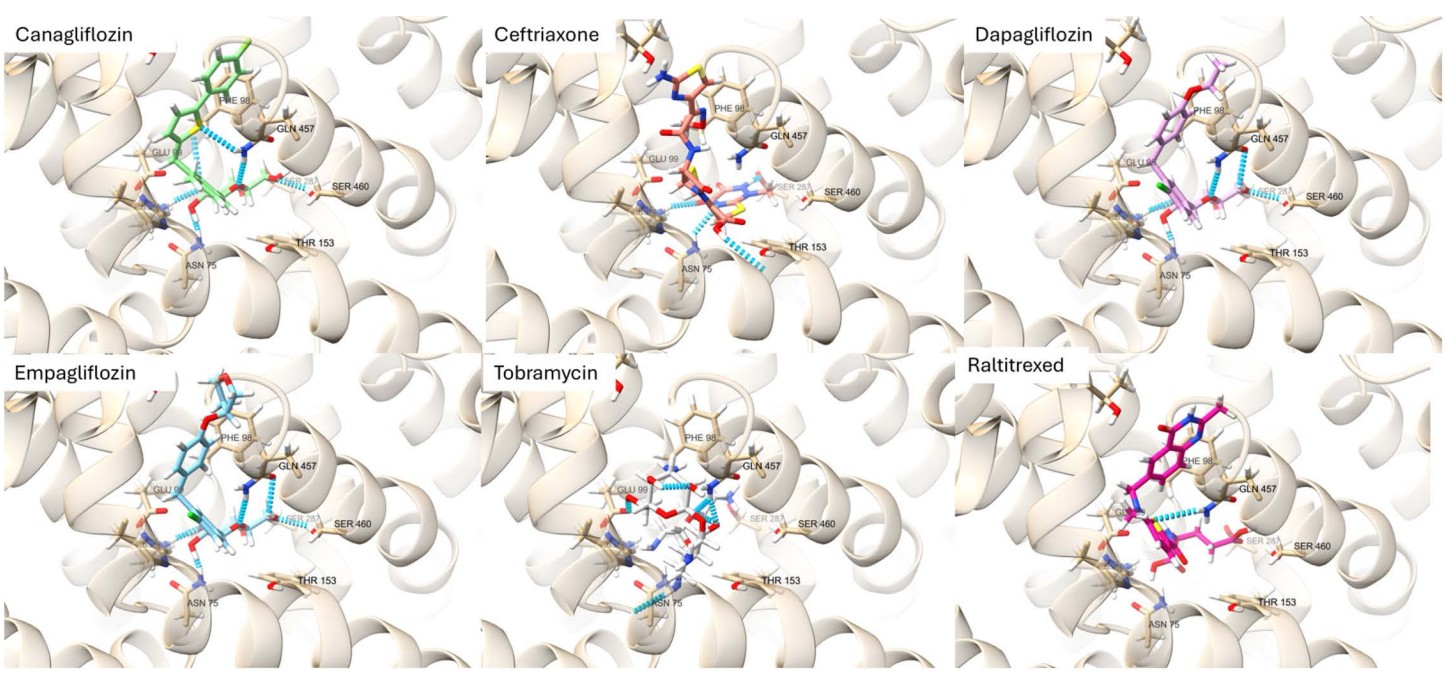

**Fig 5. The best docked conformations for the 6 best ranked compounds by RMSD in the MD simulations.**

Canagliflozin, dapagliflozin and empagliflozin exhibited similar patterns of binding with SGLT2 as shown in Fig 5. Hydrogen bonds are formed by each of these compounds and residues ASN 75 LYS321, GLN 457, and SER 460. Tobramycin and raltitrexed also both form a hydrogen bond with GLN 457. In common with the SGLT2 inhibitors ceftriaxone is additionally stabilised by hydrogen bonding to residues ASN 72, THR 321 while additional hydrogen bonds to THR 153 and SER 287 were also observed. Tobramycin formed additional hydrogen bonds with ASN 75 and GLU 99. No other

hydrogen bonds were observed in the best docked conformation of raltitrexed. Each of these molecules are relatively large drug entities and the potential for steric hinderance of glucose transport, or the binding of therapeutic drug molecules is obvious for each of the compounds displayed in Fig 5Fig:The best docked conformation of empagliflozin (A), ceftriaxone (B), ticagrelor (C) and fluvastatin (D) in the glucose channel of the SGLT2-MAP17 complex in the inward-open conformation. Hydrogen bonds are denoted by thin green lines

## Molecular Dynamics (MD)

The median RMSD over time graph in Fig 6 shows that all the tested compounds remained close to their starting conformation and below the binding deviation threshold of 5.5 Å. It is apparent from Fig 6 that some compounds have a lower median RMSD with a group of 8 compounds with values between 1 and 2 Å. This group of compounds includes the four existing SGLT2 inhibitors tested and antibiotic agents. Also noteworthy are the bound interactions with two statin compounds and one antiplatelet agent; classes of compounds frequently prescribed alongside SGLT2 inhibitors in patients with diabetes, heart failure or CKD [17].

Compounds were ranked according to the median RMSD during the last nanosecond of the simulations and these results are displayed in Table 2. Fig 6 depicts the median RMSD over the 10 nanoseconds of simulation time. All four existing licensed SGLT2 inhibitors have a median RMSD between 9 and 10 nanoseconds of less than 2 Å and are within the best seven compounds with the lowest RMSD values. Every compound examined by molecular dynamics exhibited median RMSD values lower than the 5.5 Å threshold considered to indicate whether a ligand remains bound during simulation [38].

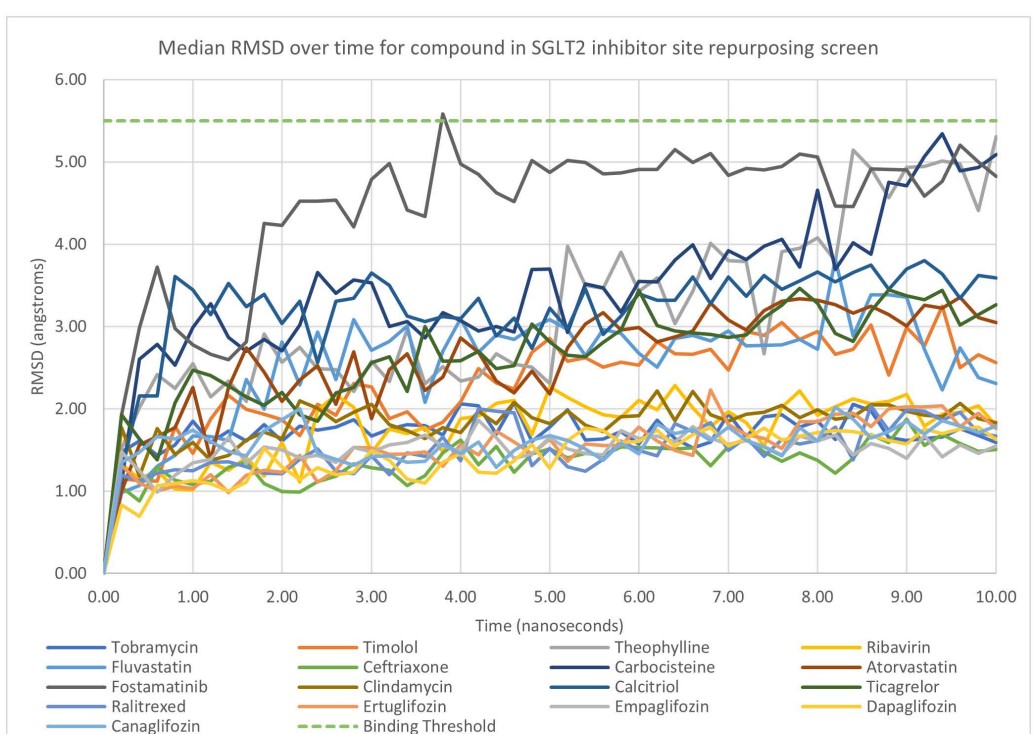

**Fig 6. Median RMSD values over time for compounds docked at the glucose/gliflozin binding site of SGLT2.**

**Table 2. Binding of compounds at the glucose/gliflozin binding site of SGLT2 ranked by their median RMSD between 9 and 10 nanoseconds. * denotes an existing SGLT2 inhibitor.**

| Name | Median RMSD 9–10 ns (Å) | Rank |
|---|---|---|
| Empagliflozin* | 1.54 | 1 |
| Ceftriaxone | 1.62 | 2 |
| Dapagliflozin* | 1.66 | 3 |
| Tobramycin | 1.69 | 4 |
| Raltitrexed | 1.78 | 5 |
| Canagliflozin* | 1.80 | 6 |
| Ertugliflozin* | 1.91 | 7 |
| Clindamycin | 1.99 | 8 |
| Ribavirin | 2.07 | 9 |
| Fluvastatin | 2.62 | 10 |
| Timolol | 2.88 | 11 |
| Atorvastatin | 3.23 | 12 |
| Ticagrelor | 3.24 | 13 |
| Calcitriol | 3.62 | 14 |
| Fostamatinib | 4.91 | 15 |
| Theophylline | 4.96 | 16 |
| Carbocisteine | 5.06 | 17 |

## Discussion

To date, most studies investigating SGLT2 inhibition have focussed on novel compounds [41]. Computational efforts to identify inhibitors have focussed on chemical homology with existing licensed inhibitors rather than investigating therapeutic compounds in current use or structural modelling with the protein [42,43].

In this study the docking results were more discriminatory than MD in terms of excluding candidate compounds. This might be because the confined space within the narrow pore represents an unfavourable environment for many compounds in docking studies, and only those compounds performing well in this docking were taken onto the MD stage.

These results show good selectivity for known binding complexes. It is not surprising that empagliflozin exhibited the lowest median RMSD value in this study as this compound was used as a reference structure in preparing the structural models and selecting a docking method. Our structural studies, in being benchmarked to the known interaction between empagliflozin and SGLT2, have deliberately focused on the selection of candidate compounds with highly similar mechanisms of interaction, improving the selectivity for likely binders in general and empagliflozin-like binders in particular. In this regard, it is pleasing to note that the other SGLT2 inhibitors were all found to have a median RMSD value of less than 2 Å in MD simulation.

Each current SGLT2 inhibitor has approximately 180 listed drug interactions in the BNF [29], however the compounds identified as likely to bind in this study do not feature prominently. Both canagliflozin and empagliflozin can increase the risk of hypotension when prescribed alongside timolol [29]. Empagliflozin has been reported to increase the risk of statin related muscle effects when given after Atorvastatin. However, the same effect is not listed for other statins or SGLT2i medication [29].

It is not clear whether the stable ligand interactions identified here would result in inhibition of sodium and glucose transport, or if the interactions could provide competitive inhibition for gliflozin compounds currently used. The strong indication of interactions with heavily used drugs likely to be prescribed alongside SGLT2 inhibitors, used currently as antibiotics, statins and antiplatelet agents, suggests that further investigation of the potential for polypharmacological

complications would be valuable. This study is limited by it considering only the protein-ligand interaction at the single site and not wider pharmacokinetic or pharmacodynamic factors.

As with all structural bioinformatics studies, the findings of this virtual library screen need validation in the form of *in vitro* studies and the analysis of clinical data. SGLT1 shares significant homology with SGLT2 with a 59.91% protein sequence similarity calculated by NCBI Blast [44,45] using the canonical sequences (P13866-1 for SGLT1 and P31639-1 for SGLT2) on the UniProt database [39]. Despite the sequence similarity between SGLT1 and SGLT2, SGLT1 is not considered a major target for existing SGLT2 inhibitors although there is a varying degree of selectivity between the drugs in this class [1,46]. Selective SGLT2 inhibitors were developed due to concerns about intestinal malabsorption [46], however a virtual screening study of SGLT1 using a library of existing medicinal compounds would be an interesting avenue for further research.

By ligand docking and molecular dynamics, we have identified compounds capable of binding near the glucose binding domain of SGLT2. These *in silico* findings require experimental corroboration but, in the meantime, the implication of several drug compounds likely to be prescribed alongside SGLT2 inhibitors such as antibiotics, statins and antiplatelet agents gives further credence to clinical concerns regarding the potential for poly-pharmacological complications in certain multimorbid patient groups.

## Author contributions

**Conceptualization:** Aled Lloyd, Karl Austin-Muttitt, Jonathan G.L. Mullins.

**Data curation:** Aled Lloyd.

**Formal analysis:** Aled Lloyd.

**Investigation:** Aled Lloyd.

**Methodology:** Aled Lloyd, Karl Austin-Muttitt, Jonathan G.L. Mullins.

**Project administration:** Aled Lloyd.

**Resources:** Aled Lloyd.

**Software:** Karl Austin-Muttitt.

**Supervision:** Karl Austin-Muttitt, Jonathan G.L. Mullins.

**Validation:** Karl Austin-Muttitt.

**Visualization:** Aled Lloyd.

**Writing – original draft:** Aled Lloyd.

**Writing – review & editing:** Aled Lloyd, Karl Austin-Muttitt, Jonathan G.L. Mullins.

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
