## [Decision Letter · Decision Letter 0]

22 May 2025

Dear Dr. Lloyd,

Thank you for submitting your manuscript to PLOS ONE. After careful consideration, we feel that it has merit but does not fully meet PLOS ONE’s publication criteria as it currently stands. Therefore, we invite you to submit a revised version of the manuscript that addresses the points raised during the review process.

We look forward to receiving your revised manuscript.

Kind regards,

Ahmed Elkamhawy

Academic Editor

PLOS ONE

Journal Requirements:

3. Thank you for uploading your study's underlying data set. Unfortunately, the repository you have noted in your Data Availability statement does not qualify as an acceptable data repository according to PLOS's standards.

Reviewers' comments:

Reviewer's Responses to Questions

**Comments to the Author**

1. Is the manuscript technically sound, and do the data support the conclusions?

Reviewer #1: Partly

Reviewer #2: Yes

2. Has the statistical analysis been performed appropriately and rigorously?

Reviewer #1: No

Reviewer #2: I Don't Know

3. Have the authors made all data underlying the findings in their manuscript fully available?

Reviewer #1: Yes

Reviewer #2: Yes

4. Is the manuscript presented in an intelligible fashion and written in standard English?

Reviewer #1: No

Reviewer #2: Yes

Reviewer #1: The manuscript, “Computational structural studies of SGLT2-related polypharmacy,” describes a computational repurposing study for the SGLT2 transporter. The topic seems to be interesting but authors have not done a great effort in discussing the problem and results of the modelling analysis in detail. Here are my comments:

1. The introduction is too short. It should give more information related to the current scenario of SGLT2 inhibitors.

2. The results part is written without sufficient details. The authors should give more details and discuss in more details.

3. The authors should discuss protein-ligand interactions more in the manuscript. Sadly, the authors did not level the residues in figure 1.

4. Molecular dynamics of 10 ns is not sufficient. It should be extended to at least 100 ns. RMSD, RMSF, RG analysis as well as MMPBSA analysis should be incorporated and discussed in detail.

5. Molecular docking and MD analysis should be compared with standard inhibitor of SGLT2 transporter.

Reviewer #2: Lloyd et al. present the results of an in vitro screen of approved, licensed compounds for their ability to interact with SGLT2, employing the ligand docking and molecular dynamics. From this screen, they identify 17 hits, including several well‐characterized SGLT2 inhibitors, and go on to discuss potential clinical interactions between these agents and other existing drugs. This work offers valuable insights for clinicians; however, we wish to raise the following concerns:

1. The authors identified 14 existing drugs, in addition to three known SGLT2 inhibitors, as compounds that interact with SGLT2. As the authors pointed out, it is important to determine whether the identified stable ligand interactions inhibit sodium and glucose transport or whether these interactions provide competitive inhibition against gliflozin. I propose in vitro analysis to clarify this point.

2. It is well established that human SGLT1 and SGLT2 share high sequence homology. Have the authors assessed whether the 17 identified compounds also interact with SGLT1 using molecular docking and molecular dynamics simulations?

3. I was unable to access the data at DOI 10.6084/m9.figshare.28363397. Could you please confirm its availability?

**Do you want your identity to be public for this peer review?** For information about this choice, including consent withdrawal, please see our Privacy Policy

Reviewer #1: **Yes: ** Dr. Shovonlal Gayen

Reviewer #2: No

---

## [Author Response · Author response to Decision Letter 1]

9 Jul 2025

Dear Ahmed Elkamhawy,

We are very grateful to you and both reviewers for reviewing our submission and for the considered feedback we have received as a result. A point-by-point reply to the comments can be found below.

Journal requirements

1. The resubmitted version of this report conforms to the PLOS file style and naming guidelines. Thank you for bringing this to our attention.

2. The modelling, docking and docking analysis tools used for this study can be found at https://proteins.swan.ac.uk/modelling-portal/. This website hosts automated structural modelling and docking services based on validated published methods freely available for academic use and credited within the manuscript. An institutional email address is required to sign up and a report detailing the tools found on this website is in press. The code used for MD simulations is freely available through github at https://github.com/bioinfkaustin/gromacs-on-colab.

3. We are sorry that the supplementary data was not available. This has now been corrected, and the files have been appropriately released for public viewing on figshare. The data files can be found at https://doi.org/10.6084/m9.figshare.28363397.v1.

Reviewer 1

We would like to thank Dr. Shovonlal Gayen for their time and for the constructive suggestions. Our responses to their comments are below.

1. We have furnished the introduction with more background information of the protein, polypharmacy as a concept and the widespread use of SGLT2 inhibitors.

2. Significant additions have been made to the results section aiming to address this concern.

3. A discussion of the protein-ligand interactions in greater detail has been included in the revised results section. We hope this addresses both this comment and comment 2 above. New figures have also been included to describe the protein structure and the ligand interactions of the 6 compounds with the lowest median RMSD in the MD studies

4. We acknowledge that through reporting only 10 nanoseconds of data, the molecular dynamics (MD) elements of this study include relatively short simulations. This important aspect of study design was not a choice taken lightly. MD studies are a valuable addition to the assessment of docking results and have been validated for this purpose1. A validation study using the DUD-E dataset identified a root mean square deviation (RMSD) threshold of 5.5Å was robust in discerning true active poses in MD simulations of 10, 50 and 100 nanosecond durations. In this study the difference in sensitivity measured by the area under a ROC curve (AUC) between each simulation length was relatively small. The calculated AUC values for experiments using the DUD-E dataset were 0.806, 0.840 and 0.836 for simulations of 10, 50 and 100 nanoseconds respectively1. The current study is the second performed using a new study protocol for high-throughput docking-MD screening that aims to make such structural bioinformatics studies achievable and accessible to new researchers to the field and researchers in training. The first, a repurposing screen of AQP1 was published in PLOS One earlier this year2. This second study of SGLT2 has a significant methodological upgrade to the AQP1 study as each ligand was simulated three times with the median RMSD value used to determine whether a ligand remained bound. There are over 50 publications within the NCBI PubMed database3–54, in addition to our recent report of the aquaporin 1 virtual screen2, which have used MD simulations of 10 nanoseconds to draw conclusions.

Despite a thorough literature review, we have been unable to identify any studies validating the use of RMSF, MMPBSA, and RG in short post-docking MD studies. Without this validation it is unclear whether these additional measurements would meaningfully enhance the study conclusions. Increasing the complexity of data analysis is contrary to the premise of the protocol under development which aims to broaden access to the field of structural bioinformatics and make studies feel more achievable to new researchers to the field.

In conclusion, we assert that three simulations of 10 nanosecond duration with assessment of ligand binding based on median RMSD are provide a validated and reproducible method to judge docked ligand conformation stability. A concise explanation of this in contained within the methods of the revised manuscript.

5. Dapagliflozin, canagliflozin, empagliflozin and ertugliflozin are licensed medicinal compounds known to act against SGLT2. These compounds have been included within all steps of the analysis through design as positive control ligands. The results for these compounds suggest stable binding with the protein. We are unclear which other “standard inhibitors” of SGLT2 reviewer 1 would like included. We cannot think of any more appropriate known inhibitors that these 4 drugs in current clinical use. We would contest that there is a large body of evidence that these compounds act against SGLT2 including experimental crystal structures confirming the interaction55–57.

We note the concern raised in the set questions regarding statistical analysis and would like clarification on whether this relates to points 3, 4, and 5, to which we have provided a robust response, or another aspect of the study.

We were disappointed to learn that this reviewer did not feel the standard of English language within the submitted manuscript met expectations. Unfortunately, no comment regarding any specific language concern or typographical errors has been made to direct improvements to the manuscript or address any perceived deficits. We have therefore concentrated on making strenuous efforts to proofread the manuscript to remove typographical errors from the revised manuscript.

Reviewer 2

We would like to thank Reviewer 2 for their time and for the constructive suggestions. Our responses to their comments are below.

1. We wholeheartedly agree that in vitro validation of these in-silico findings would be extremely valuable. We are unfortunately not able to complete these studies. Our hope in presenting this manuscript is that a laboratory with the expertise and resources to perform in vitro studies of SGLT2 would be inspired to validate these results. We have reinforced our firm belief that these findings need in vitro or clinical validation in the revised manuscript.

2. We have not studied SGLT1, mostly because despite the sequence similarity this protein is not acknowledged as a target for existing SGLT2 selective inhibitors55. However, we agree that a similar study of SGLT1 would be an interesting avenue for further research and now mention this within the discussion section of the revised manuscript.

3. The problem with the supplementary data has been addressed. Thank you for bringing this to our attention. Please see comments to the editor. The data can be found at https://doi.org/10.6084/m9.figshare.28363397.v1.

References

1. Guterres, H. & Im, W. Improving Protein-Ligand Docking Results with High-Throughput Molecular Dynamics Simulations. J. Chem. Inf. Model. 60, 2189–2198 (2020).

2. Lloyd, A. R., Austin-Muttitt, K. & Mullins, J. G. L. In silico drug repurposing at the cytoplasmic surface of human aquaporin 1. PLoS ONE 20, e0314151 (2025).

3. Agarwal, S. et al. An integrated computational approach of molecular dynamics simulations, receptor binding studies and pharmacophore mapping analysis in search of potent inhibitors against tuberculosis. J Mol Graph Model 83, 17–32 (2018).

4. Ali, M. A., Nath, A., Jannat, M. & Islam, M. M. Direct Synthesis of Diamides from Dicarboxylic Acids with Amines Using Nb2O5 as a Lewis Acid Catalyst and Molecular Docking Studies as Anticancer Agents. ACS Omega 6, 25002–25009 (2021).

5. Aliko, V. et al. ‘From shadows to shores’-quantitative analysis of CuO nanoparticle-induced apoptosis and DNA damage in fish erythrocytes: A multimodal approach combining experimental, image-based quantification, docking and molecular dynamics. Sci Total Environ 906, 167698 (2024).

6. Aloui, M. et al. QSAR modelling, molecular docking, molecular dynamic and ADMET prediction of pyrrolopyrimidine derivatives as novel Bruton’s tyrosine kinase (BTK) inhibitors. Saudi Pharm J 32, 101911 (2024).

7. Amini, R., Moradi, S., Najafi, R., Mazdeh, M. & Taherkhani, A. BACE1 Inhibition Utilizing Organic Compounds Holds Promise as a Potential Treatment for Alzheimer’s and Parkinson’s Diseases. Oxid Med Cell Longev 2024, 6654606 (2024).

8. Askari, F. S. et al. Digging for the discovery of SARS-CoV-2 nsp12 inhibitors: a pharmacophore-based and molecular dynamics simulation study. Future Virol (2022) doi:10.2217/fvl-2022-0054.

9. Azam, M. A. & Jupudi, S. Insight into the structural requirements of thiophene-3-carbonitriles-based MurF inhibitors by 3D-QSAR, molecular docking and molecular dynamics study. J Recept Signal Transduct Res 37, 522–534 (2017).

10. Azam, M. A. & Thathan, J. Pharmacophore generation, atom-based 3D-QSAR and molecular dynamics simulation analyses of pyridine-3-carboxamide-6-yl-urea analogues as potential gyrase B inhibitors. SAR QSAR Environ Res 28, 275–296 (2017).

11. Bandaru, S. et al. Molecular dynamic simulations reveal suboptimal binding of salbutamol in T164I variant of β2 adrenergic receptor. PLoS One 12, e0186666 (2017).

12. Chaudhuri, A., Bera, A. K., Sarkar, I. & Chakraborty, S. Insights from Analysis of Binding Sites of Human Meprins: Screening of Inhibitors by Molecular Dynamics Simulation Study. Comb Chem High Throughput Screen 19, 246–258 (2016).

13. Chauhan, D. et al. Design, synthesis, biological evaluation, and molecular modeling studies of rhodanine derivatives as pancreatic lipase inhibitors. Arch Pharm (Weinheim) 352, e1900029 (2019).

14. Chen, P.-Y. & Han, L.-T. Study on the molecular mechanism of anti-liver cancer effect of Evodiae fructus by network pharmacology and QSAR model. Front Chem 10, 1060500 (2022).

15. da Silva, G. D. et al. In vitro and in silico studies of the larvicidal and anticholinesterase activities of berberine and piperine alkaloids on Rhipicephalus microplus. Ticks Tick Borne Dis 12, 101643 (2021).

16. Das, B. K., Pv, P. & Chakraborty, D. Computational insights into factor affecting the potency of diaryl sulfone analogs as Escherichia coli dihydropteroate synthase inhibitors. Comput Biol Chem 78, 37–52 (2019).

17. Dhameliya, T. M., Nagar, P. R. & Gajjar, N. D. Systematic virtual screening in search of SARS CoV-2 inhibitors against spike glycoprotein: pharmacophore screening, molecular docking, ADMET analysis and MD simulations. Mol Divers 26, 2775–2792 (2022).

18. Gahtori, J., Pant, S. & Srivastava, H. K. Modeling antimalarial and antihuman African trypanosomiasis compounds: a ligand- and structure-based approaches. Mol Divers 24, 1107–1124 (2020).

19. Guerrero-Perilla, C., Bernal, F. A. & Coy-Barrera, E. Insights into the interaction and binding mode of a set of antifungal azoles as inhibitors of potential fungal enzyme-based targets. Mol Divers 22, 929–942 (2018).

20. Halder, S. K. & Elma, F. In silico identification of novel chemical compounds with antituberculosis activity for the inhibition of InhA and EthR proteins from Mycobacterium tuberculosis. J Clin Tuberc Other Mycobact Dis 24, 100246 (2021).

21. Huang, X., Dorhout Mees, E., Vos, P., Hamza, S. & Braam, B. Everything we always wanted to know about furosemide but were afraid to ask. Am J Physiol Renal Physiol 310, F958-971 (2016).

22. Jamal, S., Grover, A. & Grover, S. Machine Learning From Molecular Dynamics Trajectories to Predict Caspase-8 Inhibitors Against Alzheimer’s Disease. Front Pharmacol 10, 780 (2019).

23. Johari, S., Sharma, A., Sinha, S. & Das, A. Integrating pharmacophore mapping, virtual screening, density functional theory, molecular simulation towards the discovery of novel apolipoprotein (apoE ε4) inhibitors. Comput Biol Chem 79, 83–90 (2019).

24. Kamel, E. M. et al. Molecular modeling and DFT studies on the antioxidant activity of Centaurea scoparia flavonoids and molecular dynamics simulation of their interaction with β-lactoglobulin. RSC Adv 13, 12361–12374 (2023).

25. Kamel, E. M. et al. Xanthine Oxidase Inhibitory Activity of Euphorbia peplus L. Phenolics. Comb Chem High Throughput Screen 25, 1336–1344 (2022).

26. Kausar, M. A. et al. Identifying the alpha-glucosidase inhibitory potential of dietary phytochemicals against diabetes mellitus type 2 via molecular interactions and dynamics simulation. Cell Mol Biol (Noisy-le-grand) 67, 16–26 (2022).

27. Khan, A. A., Baildya, N., Dutta, T. & Ghosh, N. N. Inhibitory efficiency of potential drugs against SARS-CoV-2 by blocking human angiotensin converting enzyme-2: Virtual screening and molecular dynamics study. Microb Pathog 152, 104762 (2021).

28. Khan, M. K. A., Akhtar, S. & Arif, J. M. Development of In Silico Protocols to Predict Structural Insights into the Metabolic Activation Pathways of Xenobiotics. Interdiscip Sci 10, 329–345 (2018).

29. Khan, M. F. et al. Dibenzepinones, dibenzoxepines and benzosuberones based p38α MAP kinase inhibitors: Their pharmacophore modelling, 3D-QSAR and docking studies. Comput Biol Med 110, 175–185 (2019).

30. Khedr, M. A., Mohafez, O. M. M. & Al-Haider, I. A. Virtual Screening-Based Discovery of Potent Hypoglycemic Agents: In Silico, Chemical Synthesis and Biological Study. Curr Comput Aided Drug Des 16, 741–756 (2020).

31. Makeneni, S., Thieker, D. F. & Woods, R. J. Applying Pose Clustering and MD Simulations To Eliminate False Positives in Molecular Docking. J. Chem. Inf. Model. 58, 605–614 (2018).

32. Malekipour, M. H., Shirani, F., Moradi, S. & Taherkhani, A. Cinnamic acid derivatives as potential matrix metalloproteinase-9 inhibitors: molecular docking and dynamics simulations. Genomics Inform 21, e9 (2023).

33. Masumi, M. et al. Methicillin-Resistant Staphylococcus aureus: Docking-Based Virtual Screening and Molecular Dynamics Simulations to Identify Potential Penicillin-Binding Protein 2a Inhibitors from Natural Flavonoids. Int J Microbiol 2022, 9130700 (2022).

34. Modi, P., Patel, S. & Chhabria, M. Structure-based design, synthesis and biological evaluation of a newer series of pyrazolo[1,5-a]pyrimidine analogues as potential anti-tubercular agents. Bioorg Chem 87, 240–251 (2019).

35. Moussa, N., Hassan, A. & Gharaghani, S. Pharmacophore model, docking, QSAR, and molecular dynamics simulation studies of substituted cyclic imides and herbal medicines as COX-2 inhibitors. Heliyon 7, e06605 (2021).

36. Muniz Seif, E. J., Icimoto, M. Y. & Silva Júnior, P. I. In silico bioprospecting of receptors associated with the mechanism of action of Rondonin, an antifungal peptide from spider Acanthoscurria rondoniae haemolymph. In Silico Pharmacol 12, 55 (2024).

37. Nagar, P. R., Gajjar, N. D. & Dhameliya, T. M. In search of SARS CoV-2 replication inhibitors: Virtual screening, molecular dynamics simulations and ADMET analysis. J Mol Struct 1246, 131190 (2021).

38. Omoboyowa, D. A. et al. Structure-based discovery of selective CYP17A1 inhibitors for Castration-resistant prostate cancer treatment. Biol Methods Protoc 7, bpab026 (2022).

39. Omoboyowa, D. A., Balogun, T. A., Omomule, O. M. & Saibu, O. A. Identification of Terpenoids From Abrus precatorius Against Parkinson’s Disease Proteins Using In Silico Approach. Bioinform Biol Insights 15, 11779322211050757 (2021).

40. Parcha, P. et al. Identification of natural inhibitors of Bcr-Abl for the treatment of chronic myeloid leukemia. Chem Biol Drug Des 90, 596–608 (2017).

41. Patel, H. M., Shaikh, M., Ahmad, I., Lokwani, D. & Surana, S. J. BREED based de novo hybridization approach: generating novel T790M/C797S-EGFR tyrosine kinase inhibitors to overcome the problem of mutation and resistance in non small cell lung cancer (NSCLC). J Biomol Struct Dyn 39, 2838–2856 (2021).

42. Patel, H. M., Ahmad, I., Pawara, R., Shaikh, M. & Surana, S. In silico search of triple mutant T790M/C797S allosteric inhibitors to conquer acquired resistance problem in non-small cell lung cancer (NSCLC): a combined approach of structure-based virtual screening and molecular dynamics simulation. J Biomol Struct Dyn 39, 1491–1505 (2021).

43. Pola, M., Rajulapati, S. B., Potla Durthi, C., Erva, R. R. & Bhatia, M. In silico modelling and molecular dynamics simulation studies on L-Asparagi

---

## [Decision Letter · Decision Letter 1]

1 Oct 2025

Computational structural studies of SGLT2-related polypharmacy

PONE-D-25-11138R1

Dear Dr. Lloyd,

We’re pleased to inform you that your manuscript has been judged scientifically suitable for publication and will be formally accepted for publication once it meets all outstanding technical requirements.

Kind regards,

Ahmed Elkamhawy

Academic Editor

PLOS ONE

Additional Editor Comments (optional):

Reviewers' comments:

Reviewer's Responses to Questions

**Comments to the Author**

Reviewer #1: All comments have been addressed

Reviewer #2: All comments have been addressed

2. Is the manuscript technically sound, and do the data support the conclusions?

Reviewer #1: Yes

Reviewer #2: Yes

3. Has the statistical analysis been performed appropriately and rigorously?

Reviewer #1: I Don't Know

Reviewer #2: I Don't Know

4. Have the authors made all data underlying the findings in their manuscript fully available?

Reviewer #1: Yes

Reviewer #2: Yes

5. Is the manuscript presented in an intelligible fashion and written in standard English?

Reviewer #1: Yes

Reviewer #2: Yes

Reviewer #1: The manuscript may be accepted in its present form. Authors have addresssd my comments and modified the manuscript with inlcusion of new figure.

Reviewer #2: The concerns raised last time have been appropriately revised. There are no further additional comments.

**Do you want your identity to be public for this peer review?** For information about this choice, including consent withdrawal, please see our Privacy Policy

Reviewer #1: No

Reviewer #2: No

---

## [Editor Report · Acceptance letter]

PONE-D-25-11138R1

PLOS ONE

Dear Dr. Lloyd,

I'm pleased to inform you that your manuscript has been deemed suitable for publication in PLOS ONE. Congratulations! Your manuscript is now being handed over to our production team.

Kind regards,

on behalf of

Dr. Ahmed Elkamhawy

Academic Editor

PLOS ONE